# Open access publishers: The new players

**Rosângela Schwarz Rodrigues**[1], **Ernest Abadal**[2,3]*, **Breno Kricheldorf Hermes de Araújo**[4]

**1** Graduate Programa of Library and Information Science, Federal University of Santa Catarina, Florianópolis, Brazil, **2** Department of Librarianship, Information Science and Audiovisual Communication, University of Barcelona, Barcelona, Spain, **3** Research Center on Information, Communication and Culture, University of Barcelona, Barcelona, Spain, **4** Undergraduate research grantee student in Library Science, Federal University of Santa Catarina, Florianópolis, Brazil

\* abadal@ub.edu

**Data Availability Statement:** Data file are available from the Zenodo database (10.5281/zenodo.3598233).

**Funding:** Rodrigues is funded by Coordenação de Aperfeiçoamento de Pessoal de Nível Superior (CAPES Brasil) – Code 001 and Conselho Nacional

## Abstract

The essential role of journals as registries of scientific activity in all areas of knowledge justifies concern about their ownership and type of access. The purpose of this research is to analyze the main characteristics of publishers with journals that have received the DOAJ Seal. The specific objectives are a) to identify publishers and journals registered with the DOAJ Seal; b) to characterize those publishers; and c) to analyze their article processing fees. The research method involved the use of the DOAJ database, the Seal option and the following indicators: publisher, title, country, number of articles, knowledge area, article processing charges in USD, time for publication in weeks, and year of indexing in DOAJ. The results reveal a fast-rising oligopoly, dominated by Springer with 35% of the titles and PLOS with more than 20% of the articles. We've identified three models of expansion: a) a few titles with hundreds of articles; b) a high number of titles with a mix of big and small journals; and c) a high number of titles with medium-size journals. We identify a high number of titles without APCs (27%) in all areas while medicine was found to be the most expensive area. Commercial publishers clearly exercise control over the scope of journals and the creation of new titles, according to the interests of their companies, which are not necessarily the same as those of the scientific community or of society in general.

## 1 Introduction

Publishing research results in a recognized journal is the most accepted way of documenting the originality of the work and confirm that its results were good enough to overcome the skepticism of the scientific community and to be integrated into the knowledge of the area concerned. The need to publish has maintained journals as a key element of scientific research, even with all the technological and social changes of recent years [1], [2], [3], [4], [5], [6].

The essential role of journals as registries of scientific activity in all areas of knowledge justifies research on their management and the publishers that own them. The vital importance of publishing research results in terms of the prestige it gives researchers and institutions, and as a registration of copyright to a given finding for citing in subsequent studies, is a key feature of the communication of science [7], [8], [9], [10], [11].

de Desenvolvimento Científico e Tecnológico (CNPq Brasil). Abadal is funded by AGAUR, research agency of Generalitat de Catalunya (SGR 2017-422). Araujo got the Conselho Nacional de Desenvolvimento Científico e Tecnológico (CNPq Brasil) scholarship.

**Competing interests:** The authors have declared that no competing interests exist.

Mathias, Jahn and Laakso (2019, p.5) [12] remind us that "journals do not exist in a vacuum, but within a dynamic environment characterized by competition for high-quality manuscripts. Since peer-reviewed publications are still the key to academic career progression, a journal's value is closely connected to the prestige it brings to authors."

Three major commercial publishers (Elsevier, Springer-Kluwer, and Wiley-Blackwell) own 42% of all published articles and the majority of the most prestigious and widely circulated journals in 2007. Another 2,000 publishers are responsible for all the rest, none with more than 3% of the total [13]. Half of all scientific publishing is controlled by a small group of commercial publishers who offer high-priced "big deals" to libraries, despite the increasing diversity of countries of the authors of scientific production [14], perpetuating the dominance of publication by the same titles and preventing the creation of new options in open access or in peripheral countries [15], [16], [17], [18]. This situation has given rise to complaints of abusive costs and cartelization practices [19], [20], [21]. Now, according to *Ulrich's directory*, the biggest publishers of academic journals are Elsevier (The Netherlands, about 4.700 journals), SpringerNature (United Kingdom, 4.200), Wiley (United States, 3.200), Routledge (UK, 3.100), Sage (United States, 2.300) and Taylor&Francis (United Kingdom, 2.200), considering all their different companies.

In the Open Access scene, a "study of 319 journals operated by four major commercial publishers, BMC, Frontiers, MDPI, and Hindawi, indicated that higher APCs are associated with higher article volumes." (Khoo, 2019, p. 10) [22]. The author identifies a high level of APCs hyperinflation in the new publishers' prices. The study shows the replication of the ligopoly existent in the traditional publishers in the new Open Access commercial publishers.

Beasley (2016, p. 167) [8] discusses the new models of scientific publications based on Open Access, arguing that large-scale creation of titles via the so-called "gold route" can "perpetuate and even reinforce an already well-documented system of discrimination that excludes important groups from having their research disseminated through formal channels of scientific communication." Moreover, the fact that hybrid journals are already collecting APCs from authors raises the question of whether these journals should reduce their subscription prices to reflect the proportion of publication costs already paid by the author for the content [23].

Piwowar el al. (2018) [24] point out that there is significant literature on Open Access, but that the definitions are still fluid. These authors identify a large number of open access articles in a category they describe as "bronze", where the articles are free to read on the publishing institution's website but do not have a clearly identified license, which may complicate the identification of the articles as open. The definitions of types of access are still under discussion, and the authors point out that gold is one of the categories with the lowest number of articles identified, representing around 7.5% of the total sample, and with a wide range of APC values.

Piwowar, Priem and Orr (2019) [25] found that 20% of all articles were in open access in 2018, divided into four types of OA: gold, green, hybrid, and bronze. Since the focus here is on journals, we concentrate on the gold option. All the others are complementary, since the core of the change is the journals, with academic peer review and technical indexation, metadata and metrics provided by a publisher.

Aspesi et al. (2019) [26] detail the strategies of the big publishers to grow in the new scenario with open access journals and Plan S, which does not recognize either the embargo or the hybrid model as open. These commercial publishers are investing in academic management solutions using their data, creating new titles in open access and partnerships with educational and research institutions to publish their journals.

This study of the main publishers and their collections in the global scenario aims to assess concentration of ownership among the various players and countries in the Directory of Open

Access Journals (DOAJ) database. We have chosen the DOAJ because it is a representative platform for Open Access publishers and journals and it has a policy of accepting only Open Access journals, while excluding titles that choose the embargo or the so-called hybrid model. The general objective of this research is to analyze the main characteristics of publishers holding the DOAJ Seal. The specific objectives are a) to identify publishers and journals registered with the DOAJ seal; b) to characterize those publishers (number of journals, number of articles, and knowledge area); and c) to analyze their article processing fees.

## 2 Methodology

The DOAJ database was created in 2003 and includes almost 14,000 peer-reviewed open access journals covering all knowledge areas, published in 130 countries. There is a selective process to be followed to assure the quality of the titles and for this reason, DOAJ can be considered as a "white list" to confront supposed predatory publishing (i.e. a journal is not a predatory one if it's included in this directory).

DOAJ is maintained by Infrastructure Services for Open Access (IS4OA) and its funding is derived from donations (40% from publishers and 60% from the public sector). DOAJ introduced a quality distinction, called the DOAJ Seal, to identify the most prominent journals "that achieve a high level of openness, adhere to Best Practice and high publishing standards" (DOAJ) [27] (use of DOI, metadata in the articles, preservation policy, whether author holds copyright, whether reuse of content is allowed, etc.). There are 1354 journals (around 10% of the total) that have been awarded the Seal.

We chose to study the DOAJ Seal due to the large number of journals exclusively in Open Access and the rigorous criteria used to index the titles, which leaves little question of the quality of the publications. The search strategy involved using the Seal option, then ranking the journals to identify the biggest publishers, the number of journals and the number of articles in March 2019. We have extracted the following indicators from DOAJ: publisher, title, ISSN, country, number of articles, knowledge area (according to the DOAJ classification), value of article processing charges in USD, time for publication in weeks, and year of indexing in DOAJ.

For the descriptions of the publishers' characteristics we consulted their websites.

## 3 Results and discussion

The data shows a prevalence of commercial publishers, both traditional and new. We investigated both the number of titles and the number of articles, since these two categories reveal different influences on the scenario. The distribution of knowledge areas is in keeping with the relative prominence of the areas in the scientific world generally, with the highest number of titles falling into the area of medicine. Publishers and knowledge areas influence the value of APCs and the time for publication of the articles, expressed in weeks.

### 3.1 Publishers and titles registered with the DOAJ Seal

The data shows 463 publishers registered in January 2019, 123 with three titles or less and 105 with just one journal. The titles are owned predominantly by four big publishers: BioMed Central, Hindawi Limited, Multidisciplinary Digital Publishing Institute (MDPI) and Springer Open. The companies Frontier Media S.A., Copernicus Publications, African Online Scientific Information System (AOSIS), and Nature Publishing Group form a second group with significant numbers of titles. On the other hand, *PLOS One* and *International Journal of Crystallography* each stand out for a high number of articles from a single journal.

If we consider the number of articles, the data shows a different configuration. The number of articles varies widely among journals and affects the representation of the publisher in the DOAJ database and in the "market". PLOS is the leader company, with 240,000 or 20% of all articles from just seven journals. The single journal *PLOS One* has almost as many articles as the collective total for 147 smaller publishers and more than any other single publisher. Despite its small number of titles, PLOS is certainly a central player in the Open Access oligopoly of publishers, along with Bio Med Central (Springer), Hindawi Limited and Multidisciplinary Digital Publishing Institute (MDPI), each of which have more than 180,000 articles. If we add up the articles of these four publishers, they are responsible for 807,271 out of a total of 1,257,208, almost 65% of the total, representing a concentration of ownership bigger than the one described by Larivière, Hautein and Mongeon (2015) [19] in a traditional scenario.

Table 1 identifies three models of publishers in the Open Access world: a) expansion through the number of journals; b) expansion through the number of articles in few journals; and c) creation of an "open" division by a traditional publisher. The best examples of the first model are Biomed Central, MDPI and Hindawi, companies that began as digital publishers. *PLOS One*, the pioneer of the mega journal, is emblematic of the second model, beginning as a journal in 2003 and introducing two significant changes to academic publishing: an expansion of the number of articles per title, and the elimination of "novelty" from the publishing criteria: "We evaluate submitted manuscripts on the basis of methodological rigor and high ethical standards, regardless of perceived novelty" [28].

There is also what could be described as a mixed model, where a publisher with a high number of titles has a few with a much higher number of articles, but in a specific knowledge area, like Hindawi with *BioMed Research International* and *Mathematical Problems in Engineering*, and MDPI with *Sensors* and *International Journal of Molecular Sciences*. Ellers, Crowther and Harvey (2017, p. 91) [29] suggest that the higher number of articles are due to the fact that a "rejection of manuscripts is a cost factor for an open access journal with the author-pays system because rejected manuscripts need to be handled but do not generate income. This is one of the reasons why highly selective journals rarely adopt OAP, as their stringent acceptance rates would render OAP unprofitable."

Springer is responsible for 35% of the titles indexed with the DOAJ Seal, reflecting an even bigger oligopoly than the one identified in the general scientific publishing market [19], and more than the total of titles belonging to all the small publishers. The Springer case, expanding the Open Access model through acquisitions and the creation of a new company while maintaining a traditional subscription structure, is one reason for the slow progress in advancing OA, which Mathias, Jahn and Laakso (2019, p. 2) [12] describe as "an issue long discussed and recognized among all stakeholders". These authors add that "[f]urther delaying the conversion of subscription journals, or rapid adoption of new OA journals as substitutes, is the lack of incentives for publishers to accelerate this process and discard what has proven to be a highly profitable and stable business model."

Hindawi, MDPI, AOSIS and PLOS are new companies created after the Open Access movement began. Hindawi works with its own journals and in partnership with other publishers; MDPI has its own journals, partnerships with scientific societies, a preprint structure and open books; AOSIS, which is based in South Africa, publishes scholarly journals, books and educational content using Moodle.

Studies using DOIs as a source of information show a more significant number for OA because the unit of analysis is different: all DOIs, without distinguishing the period and indexation of the journals and repositories are included. This method is unsuitable if we are looking exclusively for high-quality research results. Gold Open Access titles represented only 8% of all journals in 2018 [26]. An analysis of revenues confirms the small scale of their share in the

**Table 1. Biggest publishers: Distribution of titles and articles.**

| Publishers | Titles | % | Articles | % |
|---|---|---|---|---|
| Bio Med Central (Springer) United Kingdom, 1999 | **Total** 294 | 21.3 | 183534 | 15 |
| | *BMC Public Health* | | 8639 | |
| | *BMC Genomics* | | 6695 | |
| | Others (292) | | 168200 | |
| Hindawi Limited * United Kingdom, Egypt, 1997 | **Total** 227 | 16.4 | 186847 | 15.3 |
| | *BioMed Research International* | | 15667 | |
| | *Mathematical Problems in Engineering* | | 14241 | |
| | Others (225) | | 156939 | |
| Multidisciplinary Digital Publishing Institute (MDPI) Switzerland, 1996 | **Total** 173 | 12.5 | 197265 | 16.2 |
| | *Sensors* | | 17945 | |
| | *International Journal of Molecular Sciences* | | 17123 | |
| | Others (171) | | 162167 | |
| Springer Open United Kingdom, 2010 | **Total** 170 | 12.3 | 35753 | 2.9 |
| | *Journal of High Energy Physics* | | 4180 | |
| | *Journal on Advances in Signal Processing* | | 3711 | |
| | Others (168) | | 27862 | |
| Frontiers Media S.A.** Switzerland, 2007 | **Total** 53 | 3.8 | 102732 | 8.4 |
| | *Frontiers in Psychology* | | 12674 | |
| | *Frontiers in Microbiology* | | 10933 | |
| | Others (51) | | 79125 | |
| Public Library of Science (PLOS) The United States of America, 2003 | **Total** 7 | 0.5 | 239625 | 19.6 |
| | *PLOS ONE* | | 202825 | |
| | *PLOS Genetics* | | 7908 | |
| | Others (5) | | 28892 | |
| Copernicus Publications Germany, 1988 | **Total** 38 | 2.7 | 54873 | 4.5 |
| Nature (Springer) United Kingdom, 1869 | **Total** 22 | 1.6 | 23634 | 1.9 |
| OTHERS (147) | *Acta Crystallographica* | 0.3 | 22728 | 2.2 |
| | Others (359) | 26 | 288724 | 11.7 |
| Total | 1,390 | 100 | 1,257,208 | 100 |

Data collected by the authors

* Hindawi mentions a partnership with Wiley

** Frontiers Media reports investments from Springer/Nature, although are identified as separate companies.

market: less than U$500 million in 2017, or about 5% of the total, 15 years after the Budapest Open Access Declaration was signed [25].

The growth in the number of titles from 2010 to 2019 (Table 2) varies widely among publishers: Springer Open increased its number of journals by 840%, while growth for MDPI was 540% in a media growth of 190% over an eight-year period. This reflects a clear strategy to fill the market with new titles. Springer Open, BioMed Central, and Nature Publishing are the same company, constituting a mix of spinoff publishers and mergers to operate in the open access market while maintaining the traditional publishing company.

SpringerOpen is a new brand name that retains the prestige of a traditional publisher while increasing the number of titles in open access by 800% in 8 years without ceasing or changing its activities with subscription journals. If we consider Springer also owns Nature, BioMed Central and part of Frontiers Media, it is the company with the fastest growing

Table 2. Number of titles awarded the DOAJ Seal since 2010, organized by publisher.

| Publisher | Until 2010 | 2011 | 2012 | 2013 | 2014 | 2015 | 2016 | 2017 | 2018 | Total | Growth% |
|---|---|---|---|---|---|---|---|---|---|---|---|
| BioMed Central (Springer) | 174 | 7 | 20 | 16 | 12 | 20 | 22 | 8 | 15 | 294 | 69 |
| Hindawi Limited | 76 | 15 | 17 | 31 | 9 | 18 | 27 | 20 | 14 | 227 | 200 |
| Multidisciplinary Digital Publishing Institute (MDPI) | 27 | 10 | 28 | 39 | 20 | 8 | 9 | 23 | 9 | 173 | 540 |
| SpringerOpen | 18 | 5 | 2 | 46 | 2 | 20 | 29 | 25 | 23 | 170 | 840 |
| Frontiers Media S.A. | 17 | 7 | 1 | 2 | 12 | 7 | 4 | 3 | - | 53 | 211 |
| Copernicus Publications | 18 | 4 | - | 2 | 1 | 6 | 3 | 2 | 2 | 38 | 110 |
| African Online Scientific Information Systems (AOSIS) | 10 | - | 3 | 6 | - | 1 | 9 | 5 | 1 | 35 | 250 |
| Nature Publishing Group (Springer) | 7 | - | 2 | 2 | 1 | 5 | 4 | - | 1 | 22 | 210 |
| Public Library of Science (PLOS) | 7 | - | - | - | - | - | - | - | - | 7 | 0 |
| International Union of Crystallography | 1 | - | - | - | 1 | 1 | 1 | - | - | 4 | 300 |
| Others (147) | 108 | 31 | 33 | 32 | 28 | 30 | 40 | 25 | 30 | 359 | 230 |
| **Total** | 463 | 79 | 106 | 176 | 86 | 116 | 148 | 111 | 95 | 1380 | 190 |

number of Open Access journals. This requires enormous investments and highly qualified professionals and is part of the company's strategic plan.

The year with the most significant growth was 2013, coinciding with changes to DOAJ including the introduction of more rigorous criteria for admitting new titles and the creation of the Seal [30].

## 3.2 Distribution of knowledge areas and publishers with the DOAJ Seal

Table 3 shows Medicine is the most prevalent knowledge area in the sample, with almost half of the titles and 40% of the articles, reflecting a proportion similar to the general situation of journals and articles. It is followed by science and technology with 25% of the titles and 21% of the articles. Differences in the categorization of knowledge areas among indexers makes them

Table 3. Distribution of publisher's titles and articles with the DOAJ Seal by knowledge area.

| Knowledge areas Publisher | Agriculture | | Medicine | | Science | | Technology | | Others | | 294 Total 295 | | | |
|---|---|---|---|---|---|---|---|---|---|---|---|---|---|---|
| | Art | Tit | Art | Tit | Art | Tit | Art | Tit | Art | Tit | Art | | Tit | % |
| | N | N | N | N | N | N | N | N | N | N | N | % | N | |
| African Online Scientific Information Systems (AOSIS) | 1708 | 2 | 5752 | 11 | 3435 | 3 | 2266 | 2 | 13319 | 17 | **26480** | 2.2 | **35** | 2.5 |
| BioMed Central -Springer | 6764 | 10 | 141630 | 218 | 21457 | 42 | 11765 | 15 | 1918 | 9 | **183534** | 15.0 | **294** | 21.3 |
| Copernicus Publications | 246 | 1 | - | - | 28720 | 17 | 20468 | 12 | 5439 | 8 | **54873** | 4.5 | **38** | 2,7 |
| Frontiers Media S.A | 9435 | 2 | 44849 | 24 | 32543 | 14 | 2112 | 7 | 13793 | 6 | **102732** | 8.4 | **53** | 3.8 |
| Hindawi Limited | 1557 | 7 | 86750 | 129 | 57095 | 53 | 40912 | 35 | 533 | 3 | **186847** | 15.3 | **227** | 16.4 |
| International Union of Crystallography | - | - | - | - | 26799 | 4 | - | - | - | - | **26799** | 2.2 | **4** | 0,3 |
| Multidisciplinary Digital Publishing Institute (MDPI) | 2841 | 6 | 23252 | 29 | 79068 | 57 | 81028 | 53 | 11076 | 28 | **197265** | 16.2 | **173** | 12.5 |
| Nature Publishing Group—Springer | - | - | 22895 | 9 | 404 | 6 | 319 | 6 | 16 | 1 | **23634** | 1.9 | **22** | 1.6 |
| Public Library of Science (PLOS) | - | - | 220580 | 4 | 19045 | 3 | - | - | - | - | **239625** | 19.6 | **7** | 0.5 |
| SpringerOpen | 520 | 6 | 2533 | 31 | 14124 | 29 | 14300 | 61 | 4276 | 43 | **35753** | 2.9 | **170** | 12.3 |
| Others (147) | 3517 | 8 | 30975 | 74 | 29509 | 60 | 8404 | 20 | 70471 | 197 | **142876** | 11.7 | **359** | 25.9 |
| Total | 26588 | 42 | 579216 | 529 | 312199 | 288 | 181574 | 211 | 120841 | 312 | 1220418 | 100 | 1382 | 100 |
| % | 2.2 | 3 | 47.5 | 38 | 25,6 | 21 | 15 | 15 | 10 | 22 | 100 | - | 100 | - |

Data collected by the authors. Art = articles, Tit = titles

difficult to compare, but medicine is invariably the biggest area in every study of scientific journals and articles. This predominance is also found among publishers that already have a tradition in the area and have migrated their journals to the Open Access option, like BMC and Nature. However, new players like MBPI and AOSIS show a more balanced representation of areas.

SpringerOpen has a remarkably small number of titles in the field of medicine, focusing instead on science and technology. This is probably due to Springer's merger with Nature, which is already a major player in medicine. It is important to note the impressive growth of new titles from Springer with consequences that can affect to the scientific community (to publish a huge number of journals is relevant for controlling knowledge areas). It also raises the question of how a company that is creating such an impressive number of new journals could be meeting the challenge of publishing enough good quality articles in their first years before they can be indexed in databases like WoS or Scopus. The wave of "predatory lists" of open access publishers in recent years, which have called into question the reputability of many new journals, especially from peripheral countries, is also relevant to this issue, despite the fragilities (not reliable, nor transparent, have flaws and need to be updated constantly) of these lists [31], [32].

Another option for growth is through partnerships with societies, as described by Mathias, Jahn and Laakso (2019, p. 22) [12]: "Through publishing partnerships, societies gain access to technical infrastructure and marketing resources, but are also subject to the publisher's policies and regulations. In particular, the inclusion in a journal package can have positive and negative implications, as discounted subscription fees of 'big deals' can affect the societies' revenue share."

Partnerships between commercial publishers and non-commercial institutions are a common practice that complicates classification and analysis. According to Crawford (2018, p. 20) [33], "there are 13 publisher names in the APCLand group (Springer, Nature and BioMed Central are listed separately in DOAJ) and one anomaly: because of its large stable of society-sponsored journals, Elsevier appears to have published more no-fee than fee 2017 articles in gold OA journals (a few more: 899 out of nearly 30,000)."

The high concentration of journals controlled by commercial publishers may generate an oligopoly equivalent at the one in the subscription model, since the concentration by commercial publishers in DOAJ Seal replicates in small scale the traditional publishing market [22]. This is more obvious in "core" areas like medicine and science and technology; it seems that the number of humanities journals was comparatively low in the traditional print era and are is similarly small in the open digital publishing era, or that financial interest is simply lower in the humanities. Small publishers cover the widest diversity of areas.

## 3.3 APCs charged by journals and publishers with the DOAJ Seal

Table 4 shows that 28% of journals don't charge APCs and our analysis found that small publishers, probably operated by associations and universities, are more likely not to charge APCs (although there is almost 50% of SpringerOpen titles with no APCs).

Table 4 confirms Holley's assertion (2018, p. 236) [23] that

"[t]he most critical development in publishing has been the ability of large commercial publishers to find ways to profit from open access. While the original intent of open access was to limit or destroy their monopoly, the exact opposite has happened. They have created a new revenue stream from gold APCs while still mostly retaining their subscriptions for paywalled publications, even in hybrid journals.

Table 4. Number of titles by publishers and APCs in USD.

| Publisher | No APC | | 0–500 | | 501–1000 | | 1001–1500 | | 1501–2000 | | >2001 | | Total | |
|---|---|---|---|---|---|---|---|---|---|---|---|---|---|---|
| | N | % | N | % | N | % | N | % | N | % | N | % | N | % |
| African Online Scientific Information Systems (AOSIS) | 12 | 0.9 | 23 | 1.6 | - | - | - | - | - | - | - | - | 35 | 2.5 |
| BioMed Central (Springer) | 17 | 1.2 | - | - | 9 | 0.6 | 12 | 0.9 | 245 | 17.7 | 11 | 0.8 | 294 | 21.3 |
| Copernicus Publications | 10 | 0.7 | 5 | 0.4 | 8 | 0.6 | 15 | 1.1 | - | - | - | - | 38 | 2.7 |
| Frontiers Media S.A | - | - | - | - | 3 | 0.2 | 7 | 0.5 | 20 | 1.5 | 23 | 1.7 | 53 | 3.8 |
| Hindawi Limited | 2 | 0.5 | 31 | 2.2 | 135 | 9.8 | 36 | 2.6 | 22 | 1.6 | 1 | 0.1 | 227 | 16.4 |
| International Crystallography | - | - | 3 | 0,2 | - | - | 1 | 0.1 | - | - | - | - | 4 | 0.3 |
| Multidisciplinary Digital Publishing Institute (MDPI) | 48 | 3.5 | 50 | 3.6 | 40 | 2.9 | 20 | 1.5 | 15 | 1.1 | - | - | 173 | 12.5 |
| Nature Publishing Group (Springer) | - | - | - | - | - | - | - | - | 4 | 0.3 | 18 | 1.3 | 22 | 1.6 |
| Public Library of Science (PLOS) | - | - | - | - | - | - | 1 | 0.1 | - | - | 6 | 0.4 | 7 | 0.5 |
| SpringerOpen | 81 | 5.9 | - | - | 32 | 2.3 | 42 | 3 | 14 | 1 | 1 | 0.1 | 170 | 12.3 |
| Others (147) | 208 | 15 | 66 | 4.8 | 38 | 2.7 | 32 | 2.3 | 8 | 0.6 | 7 | 0.5 | 359 | 26 |
| Total | 378 | 27.6 | 178 | 12,9 | 265 | 19.2 | 166 | 12 | 328 | 23.7 | 67 | 4.8 | 1382 | 100 |

Data collected by the authors

The complexity of adding open access to the scholarly communication system has induced much smaller scholarly and other publishers to agree to be partners by these large publishers, which has only increased their competitive dominance. Without some major unexpected change, open access, paywalled, and hybrid journals will coexist for the near future."

59% of journals charge APCs of less than US$1,000, confirming that it is possible to have quality journals charging reasonable fees. The high number of titles with no fees in important areas requires a more detailed analysis, preferably a longitudinal one, because APCs may be introduced by journals as they evolve and their position in the rankings changes. Mathias, Jahn and Laakso (2019, p. 22) [12] comment that

"commercial publishers have generally been slow in flipping subscription journals to OA at a larger scale, and, more notably, have instead resorted to creating new OA journals, or acquiring established OA journals and entire publishers (e.g., BioMed Central, Co-Action, Dove Medical Press, Medknow). Another expression of this reluctant stance to flipping is the emergence of so-called mirror journals: new fully OA journals that capitalize on existing subscription journals. Instead of converting the subscription journal to OA, a separate OA version runs alongside it, sharing the same aims and scope, almost the same name, an identical editorial board, and the same submission system (e.g., the newly founded Research Policy X and Water Research X journals by Elsevier). While not technically hybrid OA anymore, the strategic function of mirror journals for publishers appears similar: retaining the subscription-based core of the business while selling optional OA and potentially circumventing the hybrid ban that has become part of some funding policies."

For commercial publishers, revenues from subscription journals are much more significant than what they receive from Open Access. According to SPARC (2019, p. 25) [25] revenues from Springer's Nature journals division are 1,164,400,000 euros and revenues from articles are 4,386 euros. The proportions are similar for Elsevier and Wiley. Lawson, Gray and Mauri's suggestion (2016, p. 25) [34] of a "systemic opacity both within institutions as well as regarding the 'black box' of finances around scholarly communication in the UK as a whole" can be applied to the global scenario. Since revenues from open access articles are so insignificant compared to subscriptions for commercial publishers, there is no incentive to abandon traditional subscription journals while they continue to be profitable, even though the value of the

open option is not small. Table 4 shows that more than 28% of titles charge more than US $1,500 per article, and almost 5% charge more than US$2,000 per article published. The highest prices are imposed by Springer Nature, with 80% of its journals charging more than US $2,000 an article, and Springer's BioMed Central, with 87% of its titles charging more than US $1,500. SpringerOpen, has no charges for 47% of its titles, reflecting a very different price strategy.

A connection can be identified between these results and the age of the titles presented in Table 1. Springer's BioMed Central had the highest number of titles in 2010 and is a consolidated publisher that has been growing constantly since that year. Springer Nature is also a traditional publisher that has tripled the number of titles it has with the DOAJ Seal since 2010. The more recent titles come from SpringerOpen, which charges little or nothing for most of its new journals, probably just until they are consolidated in the "market". It is important to note that there is no control over the prices that journals can charge.

There are two possible explanations for commercial publishers making their titles in Open Access free of charge: a) they are in partnerships with a scientific society that pays the costs; or b) the titles are free until they are indexed in a relevant database and/or have achieved a reasonable level of prominence in the area, at which time fees may be introduced. This is an established strategy to conquer and control new markets and that is being used in various areas [35]. New studies of this situation are needed to clarify these practices.

The relationship between knowledge areas and APC amounts was analyzed to identify possible patterns (Table 5). The most surprising result was the percentage of free APC titles, totaling 27% in all areas. The least represented areas are humanities and social sciences, although the use by DOAJ of the Knowledge Areas distribution from the American Library difficult a global analysis, since they do not have equivalence with other classifications.

The area with the most expensive titles is medicine, with around 50% of titles charging more than US$1,500 to publish an article. Science and technology follows with a more balanced range of charges. All areas have a few titles with no charges. However, in education, language and literature, and social sciences most titles have no fees and just a few charges more than US$1,000 per article. It is important to remember, however, that a journal may change its APC policy as it evolves, as its impact grows.

APC pricing is connected with the publisher's policy and interests. The creation of new marketing models affects all prices. Springer's new Read and Publish (RAP) option, which combines OA publishing in hybrid journals with access to subscription content in one agreement and one fee customized for each institution [36] or country, will add even more complexity and dependence to already opaque pricing practices and dominated by oligopolies market.

The time required for publication is an increasing concern for scientists, who depend on articles to document their research. Publication is a crucial part of a researcher's career and, as Ellers, Crowther and Harvey (2017, p. 97) [29] note, "[a]s increased pressure to publish is a general pattern in academia, it makes the high acceptance rates and rapid review system of mega-journals increasingly attractive."

Rapid publication shortens the time for peer review, which may raise concerns about the rigor of the review, as shown in Table 6. Our analysis of the time for publication among DOAJ Seal journals reveals considerable diversity. The shortest publication time in the sample is 5 weeks, which may be considered a very short time to complete the whole review process. The biggest proportion of titles (32%) are in the 11–13 week range. Journals with the most expensive APCs take 14 weeks to publish an article while the cheapest journals (under 500$) take around 10 to 13 weeks. The longest time we considered relevant is 30 weeks; a total of 61 titles

**Table 5. Article Processing Charges of Journals with the DOAJ Seal by knowledge area (in USD).**

| Knowledge area | No APC | | 1–500 | | 501–1000 | | 1001–1500 | | 1501–2000 | | >2001 | | Total | |
|---|---|---|---|---|---|---|---|---|---|---|---|---|---|---|
| | N | % | N | % | N | % | N | % | N | % | N | % | N | % |
| Agriculture | 11 | 0.8 | 5 | 0.4 | 12 | 0.9 | 3 | 0.2 | 9 | 0.6 | 2 | 0.2 | 42 | 3.1 |
| Auxiliary History | 3 | 0.2 | 5 | 0.4 | 0 | 0.0 | 1 | 0.1 | 0 | 0.0 | 0 | 0.0 | 9 | 0.6 |
| Bibliography/Library | 10 | 0.7 | | 0.0 | | 0.0 | | 0.0 | | 0.0 | | 0.0 | 10 | 0.7 |
| Education | 28 | 2.0 | 4 | 0.3 | 5 | 0.4 | 3 | 0.2 | 2 | 0.2 | 1 | 0.1 | 43 | 3.1 |
| Fine Arts | 14 | 1.0 | 1 | 01 | 3 | 0.2 | 1 | 0.1 | | 0.0 | | 0.0 | 19 | 1.4 |
| General Works | 7 | 0.5 | 4 | 0.3 | 2 | 0.2 | | 0.0 | 2 | 0.4 | | 0.0 | 15 | 1.1 |
| Geography/Anthropology | 18 | 1.3 | 6 | 0.4 | 8 | 0.6 | 7 | 0.5 | 3 | 0.2 | | 0.0 | 42 | 3.1 |
| History | 10 | 0.7 | 1 | 0.1 | | 0.0 | | 0.0 | | 0.0 | | 0.0 | 11 | 0.8 |
| American History | 6 | 0.4 | | 0.0 | | 0.0 | | 0.0 | | 0.0 | | 0.0 | 6 | 0.4 |
| Language and Literature | 24 | 1.7 | 5 | 0.4 | 1 | 0.1 | 1 | 0.1 | | 0.0 | | 0.0 | 31 | 2.2 |
| Law | 11 | 0.8 | 2 | 0.2 | | 0.0 | | 0.0 | | 0.0 | | 0.0 | 13 | 1.0 |
| Medicine | 62 | 4.5 | 52 | 3.8 | 113 | 8.2 | 45 | 3.3 | 196 | 14.2 | 61 | 4.4 | 529 | 38.3 |
| Music | 2 | 0.2 | | 0.0 | | 0.0 | | 0.0 | | 0.0 | | 0.0 | 2 | 0.2 |
| Naval Science | | 0.0 | | 0.0 | 1 | 0.1 | | 0.0 | | 0.0 | | 0.0 | 1 | 0.1 |
| Philosophy/Psychology/Religion | 13 | 0.9 | 8 | 0.6 | 4 | 0.3 | 3 | 0.2 | 1 | 0.1 | 2 | 0.2 | 31 | 2.2 |
| Political Science | 10 | 0.7 | 2 | 0.2 | 1 | 0.1 | 1 | 0.1 | | 0.0 | | 0.0 | 14 | 1.0 |
| Science | 48 | 3.5 | 48 | 3.5 | 65 | 4.7 | 50 | 3.6 | 52 | 3.8 | 25 | 1.8 | 288 | 20.8 |
| Social Sciences | 37 | 2.7 | 14 | 1.0 | 9 | 0.6 | 6 | 0.4 | | 0.0 | | 0.0 | 66 | 4.8 |
| Technology | 64 | 4.6 | 21 | 1.5 | 46 | 3.3 | 45 | 3.2 | 28 | 2.0 | 7 | 0.5 | 211 | 15.3 |
| Total | 378 | 27.3 | 178 | 12.8 | 270 | 19.5 | 166 | 12.0 | 293 | 21.2 | 98 | 7.1 | 1383 | 100 |

Data collected by the author

had publication times longer than 31 weeks, representing 4.4% of the total. We were unable to identify any causal relationship between APC price and time for publication.

## 4 Conclusion

Our analysis of journals registered with the DOAJ Seal reveals a remarkable concentration of ownership. The four biggest commercial publishers are responsible for 63% of the titles indexed with the DOAJ Seal. If we add together the figures for all publishers owned by Springer (BioMed Central, SpringerOpen, and Nature), we find 35% of journals and 65% of articles in just one company. *PLOS One* alone has more articles than all the small publishers put together. If we consider the other commercial publishers that have titles in OA, the concentration of the oligopoly is even denser than the general publishing market. The concentration replicates in small scale the traditional publishing market, where just 8% of the titles are in Gold Open Access. Since it is the owners of the journals that determine the creation of new titles and the acceptance policies for papers in each knowledge area, it is reasonable to conclude that academia has little control over the scope of the journals or the creation of new titles, as these are subject to the diverse interests of commercial publishers.

In relation to this oligopoly, Larivière, Haustein and Mongeon (2015, p. 108) [19] conclude that "the role of universities and research councils cannot be over-emphasized, as they are at the heart of the research evaluation system and decide what has value. Should they create incentives for scholars to publish in open access, not-for-profit journals—rather than focusing on Impact Factors or university rankings, which clearly favor big publishers—the research community could regain control of the scholarly communication system."

**Table 6. DOAJ Seal journal times for publication (in weeks) and APC values (in USD).**

| Time weeks | APC values in US$ | | | | | | | | | |
| --- | --- | --- | --- | --- | --- | --- | --- | --- | --- | --- |
| | 0–500 | | 501–1000 | | 1001–1500 | | 1501 > | | Total | |
| | N | % | N | % | N | % | N | % | N | % |
| 5 or less | 24 | 1.01 | 7 | 0.36 | 5 | 0.29 | 8 | 0.51 | 44 | 2.17 |
| 6 | 32 | 2.32 | 7 | 0.51 | 5 | 0.36 | 10 | 0.72 | 54 | 3.91 |
| 7 | 9 | 0.65 | 2 | 0.14 | - | - | 8 | 0.58 | 19 | 1.37 |
| 8 | 21 | 1.52 | 3 | 0.22 | 4 | 0.29 | 14 | 1.01 | 42 | 3.04 |
| 9 | 6 | 0.43 | - | - | 3 | 0.22 | 12 | 0.87 | 21 | 1.52 |
| 10 | 40 | 2.89 | 17 | 1.23 | 4 | 0.29 | 12 | 0.87 | 73 | 5.28 |
| 11 | 76 | 5.50 | 47 | 3.40 | 22 | 1.59 | 26 | 1.88 | 171 | 12.37 |
| 12 | 65 | 4.70 | 7 | 0.51 | 12 | 0.87 | 33 | 2.39 | 117 | 8.47 |
| 13 | 67 | 4.85 | 35 | 2.53 | 34 | 2.46 | 29 | 2.10 | 165 | 11.94 |
| 14 | 9 | 0.65 | 20 | 1.45 | 10 | 0.72 | 44 | 3.18 | 83 | 6.01 |
| 15 | 24 | 1.74 | 13 | 0.94 | 14 | 1.01 | 14 | 1.01 | 65 | 4.70 |
| 16 | 27 | 1.95 | 11 | 0.80 | 9 | 0.65 | 21 | 1.52 | 68 | 4.92 |
| 17 | 8 | 0.58 | 20 | 1.45 | 3 | 0.22 | 17 | 1.23 | 48 | 3.47 |
| 18 | 3 | 0.22 | 21 | 1.52 | 6 | 0.43 | 16 | 1.16 | 46 | 3.33 |
| 19 | 9 | 0.65 | 10 | 0.72 | 4 | 0.29 | 17 | 1.23 | 40 | 2.89 |
| 20 | 35 | 2.53 | 10 | 0.72 | 2 | 0.14 | 19 | 1.37 | 66 | 4.78 |
| 21 | 6 | 0.43 | 13 | 0.94 | 1 | 0.07 | 12 | 0.87 | 32 | 2.32 |
| 22 | 7 | 0.51 | 4 | 0.29 | 3 | 0.22 | 9 | 0.65 | 23 | 1.66 |
| 23 | 3 | 0.22 | 3 | 0.22 | 1 | 0.07 | 16 | 1.16 | 23 | 1.66 |
| 24 | 13 | 0.94 | 2 | 0.14 | 2 | 0.14 | 14 | 1.01 | 31 | 2.24 |
| 25 | 5 | 0.36 | 2 | 0.14 | 1 | 0.07 | 5 | 0.36 | 13 | 0.94 |
| 26 | 8 | 0.58 | 2 | 0.14 | - | - | 5 | 0.36 | 15 | 1.09 |
| 27 | 2 | 0.14 | 1 | 0.07 | - | - | 4 | 0.29 | 7 | 0.51 |
| 28 | 1 | 0.07 | 5 | 0.36 | 3 | 0.22 | 12 | 0.87 | 21 | 1.52 |
| 29 | 3 | 0.22 | - | - | - | - | - | - | 3 | 0.22 |
| 30 | 14 | 1.01 | 3 | 0.22 | 1 | 0.07 | 3 | 0.22 | 21 | 1.52 |
| 31 or more | 39 | 2.8 | 4 | 0.3 | 17 | 1.2 | 11 | 0.8 | 61 | 4.4 |
| **Total** | 556 | 40.23 | 269 | 19.46 | 166 | 12.01 | 391 | 28.29 | 1380 | 100 |

Fyfe et al. (2017, p. 19) [37] are clear about the solution to this situation: "universities and learned societies are the key institutions that reward academics and should have an active role to play in creating a non-profit, online model for academic publishing that meets academic desires both to circulate and share knowledge widely and to gain prestige among peers. They could do this by offering direct support for non-profit publishers (which deliver better value for money), or by harnessing emerging technologies to establish their own publishing venues."

The identification of a wide range of APC amounts and a correlation of those amounts with the age of the titles of DOAJ seal but not with the knowledge areas or the time for publication raises questions about the strategies for the creation of new journals and their consequences for the scientific community in the long term. The results allow us to conclude that there is an oligopoly of commercial publishers trying to control the scientific communication system, creating a level of dependence where researchers have little power to decide what and where to publish since their institutions expect publications in journals with high Impact Factors that are not necessarily the best ones to dialogue with their peers, or they may have"deals" with certain journals in which their researchers are expected to publish their work. This interference in

researchers' decisions of where to publish undermines the freedom and autonomy of science, quite apart from the already well-known problem of abusive prices.

## Author Contributions

**Conceptualization:** Rosângela Schwarz Rodrigues, Ernest Abadal, Breno Kricheldorf Hermes de Araújo.

**Data curation:** Rosângela Schwarz Rodrigues, Breno Kricheldorf Hermes de Araújo.

**Investigation:** Rosângela Schwarz Rodrigues.

**Methodology:** Rosângela Schwarz Rodrigues, Ernest Abadal, Breno Kricheldorf Hermes de Araújo.

**Validation:** Rosângela Schwarz Rodrigues.

**Writing – original draft:** Rosângela Schwarz Rodrigues, Ernest Abadal, Breno Kricheldorf Hermes de Araújo.

**Writing – review & editing:** Rosângela Schwarz Rodrigues, Ernest Abadal.

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
