## [Decision Letter · Decision Letter 0]

19 Feb 2020

PONE-D-20-00381

Open access publishers: the DOAJ Seal profile

PLOS ONE

Dear Dr Abadal,

Thank you for submitting your manuscript to PLOS ONE. After careful consideration, we feel that it has merit but does not fully meet PLOS ONE’s publication criteria as it currently stands. Therefore, we invite you to submit a revised version of the manuscript that addresses the points raised during the review process.

Your manuscript has been scientifically judged by two acknowledged experts in the area. Both them have remarked the overall merit of the submitted work, and (based in my own assessment) I should say that this paper has a good potential for being considered as publishable in PLOS ONE. Nevertheless, you will find some comments (below) suggesting many improvements (some of them major) that need to be addressed through a revision of the paper.

Please perform a careful and complete revision of all these queries, comments and suggestion, and try to support as good as possible the statements contained in your rebuttal letter (and in the revised text), detailing all the modifications and improvements done.

We would appreciate receiving your revised manuscript by Apr 04 2020 11:59PM. To enhance the reproducibility of your results, we recommend that if applicable you deposit your laboratory protocols in protocols.io, where a protocol can be assigned its own identifier (DOI) such that it can be cited independently in the future. For instructions see: http://journals.plos.org/plosone/s/submission-guidelines#loc-laboratory-protocols

We look forward to receiving your revised manuscript.

Kind regards,

Sergio A. Useche, Ph.D.

Academic Editor

PLOS ONE

Journal Requirements:

1. Please consider changing the title so as to meet our title format requirement (https://journals.plos.org/plosone/s/submission-guidelines). In particular, the title should be "Specific, descriptive, concise, and comprehensible to readers outside the field" and in this case it is not informative and specific about your study's scope and methodology.

2. We note you have included tables to which you do not refer in the text of your manuscript. Please ensure that you refer to Tables 2,3,5,6 in your text; if accepted, production will need this reference to link the reader to the Table.

Reviewers' comments:

Reviewer's Responses to Questions

**Comments to the Author**

1. Is the manuscript technically sound, and do the data support the conclusions?

Reviewer #1: Partly

Reviewer #2: Yes

2. Has the statistical analysis been performed appropriately and rigorously? 

Reviewer #1: Yes

Reviewer #2: Yes

3. Have the authors made all data underlying the findings in their manuscript fully available?

Reviewer #1: Yes

Reviewer #2: Yes

4. Is the manuscript presented in an intelligible fashion and written in standard English?

Reviewer #1: Yes

Reviewer #2: Yes

5. Review Comments to the Author

Reviewer #1: There is some interesting analysis here that adds to the understanding of open access and DOAJ’s role in helping identify high quality OA journals. The paper seems incomplete, however. Additional analysis, substantiation of several claims, and expanding the description of publishers would make this a better paper. In addition, the introduction to DOAJ should be expanded for those readers who are not familiar with it. See some details below.

Lines 176, 177, 178. This statement is confusing and unsubstantiated. I suggest deleting it. You can just end the paragraph without it or find and cite a source that offers a substantiated version of what you are trying to say about novelty. Equating lack of novelty with higher quality and duplication confuses issues as it is stated. If the statement is rewritten and a source cited that supports it, note that “brings” should be “bring” in line 177.

In the introduction you mention that 3 large commercial publishers dominate the publishing industry/number of titles. In the findings you mention that PLOS and Biomed Central, MDPI and Hindawi in addition to Springer dominate the findings. You need to describe these publishers more and earlier in terms of commercial or non-profit, subject matter, etc. I would suggest expanding the introductory section on publishers to give more details rather than just those top 3 so the publishers you mention in the findings are not completely new to the reader when we reach this point. Perhaps a table of publishers would help. (For example, an extension of Table 1 in https://www.stm-assoc.org/2018_10_04_STM_Report_2018.pdf

Lines 200+ you mention some characteristics, but information about publishers is scattered, so this introduction would help readers put these scattered statements into context.

Your Table 1: should be titled “…by the Biggest Open Access Publishers”

It seems that Table 2 should logically come before Table 1.

Lines 257-260: This comment is unsubstantiated and should be supported or deleted. It is jarring and seemingly irrelevant. 257 “This highlights what appears

258 to be the real motivation behind the creation of new journals in Open Access: the

259 company’s portfolio interests in the market, rather than the needs of society or the

260 pursuit of scientific innovation.”

Line 269: Also see: Shen, C., & Björk, B.-C. (2015). ‘Predatory’ open access: a longitudinal study

of article volumes and market characteristics. BMC Medicine, 13(230), 1-15. doi:

10.1186/s12916-015-0469-2

You should expand the discussion of predatory OA and how DOAJ hopes to provide a “white list” that eliminates those.

Since in your conclusion (lines 283-285) you make a bold prediction (283: “The high concentration of journals controlled by commercial publishers may

284 generate a price crisis in the Open Access movement similar to that of the 1960s, when

285 prices rose so high that libraries were forced to cancel subscriptions.”) you need to include analysis on prices. This statement again needs evidence and your study is in a position to do so, or at least include references and a discussion of price studies by others.

In summary: include characteristics of publishers in introductory section, remove or substantiate several statements that are not substantiated by evidence from data analysis or literature review, add to the literature and discussion on predatory journals and how DOAJ serves as a white list to help authors select reputable OA journals.

Reviewer #2: This manuscript represents an interesting analysis of the publishers business model and journals' behaviour respect to some critical variables in the OA era. Focussing on the DOAJ seal is a good point of analysis. The overall manuscript is well-written, the aims of the study and the conclusions are consistent.

just a few observations:

2.Methodology section: when you give numbers, please specify the date of your search as nowaday the numbers are different, of course, even if slightly.

3.1: the observation of lines 210-212 deserves to be stressed also in the conclusion section because is very meaningful!

3.2 Distribution of knowlege areas, line 257-260: this statement is more a consideration than a description. you could put it in the Results and Discussion section.

Just a final observation: discussing about PLOSone in PLOSone...I think your analysis is quite objective but...

in conclusion, it's a manuscript, very interesting to be read!

6. PLOS authors have the option to publish the peer review history of their article (what does this mean?). If published, this will include your full peer review and any attached files.

Reviewer #1: No

Reviewer #2: No

---

## [Author Response · Author response to Decision Letter 0]

24 Mar 2020

PONE-D-20-00381 - Open access publishers: the DOAJ Seal profile

Response to reviewers

1. Please consider changing the title so as to meet our title format requirement (https://journals.plos.org/plosone/s/submission-guidelines). In particular, the title should be "Specific, descriptive, concise, and comprehensible to readers outside the field" and in this case it is not informative and specific about your study's scope and methodology.

We have changed it. We propose three options but we are open to new suggestions coming from the reviewers or the editors.

2. We note you have included tables to which you do not refer in the text of your manuscript. Please ensure that you refer to Tables 2,3,5,6 in your text; if accepted, production will need this reference to link the reader to the Table.

Revised.

Reviewer #1

There is some interesting analysis here that adds to the understanding of open access and DOAJ’s role in helping identify high quality OA journals. The paper seems incomplete, however. Additional analysis, substantiation of several claims, and expanding the description of publishers would make this a better paper.

Ok, we answer your comments below.

In addition, the introduction to DOAJ should be expanded for those readers who are not familiar with it. See some details below.

We have added more details about DOAJ (line 113).

Lines 176, 177, 178. This statement is confusing and unsubstantiated. I suggest deleting it. You can just end the paragraph without it or find and cite a source that offers a substantiated version of what you are trying to say about novelty. Equating lack of novelty with higher quality and duplication confuses issues as it is stated. If the statement is rewritten and a source cited that supports it, note that “brings” should be “bring” in line 177.

Deleted.

In the introduction you mention that 3 large commercial publishers dominate the publishing industry/number of titles. In the findings you mention that PLOS and Biomed Central, MDPI and Hindawi in addition to Springer dominate the findings. You need to describe these publishers more and earlier in terms of commercial or non-profit, subject matter, etc. I would suggest expanding the introductory section on publishers to give more details rather than just those top 3 so the publishers you mention in the findings are not completely new to the reader when we reach this point. Perhaps a table of publishers would help. (For example, an extension of Table 1 in https://www.stm-assoc.org/2018_10_04_STM_Report_2018.pdf

We have included a list of the current biggest academic publishers according to Ulrich’s and a new paragraph analyzing the most prominent open access publishers (line 71).

Lines 200+ you mention some characteristics, but information about publishers is scattered, so this introduction would help readers put these scattered statements into context.

We agree on that, thanks.

Your Table 1: should be titled “…by the Biggest Open Access Publishers”

Added.

It seems that Table 2 should logically come before Table 1.

We do not agree with that. We think that table 1 is more general and offers a global overview of the publisher’s scenario. Table 2 includes just the years.

Lines 257-260: This comment is unsubstantiated and should be supported or deleted. It is jarring and seemingly irrelevant. 257 “This highlights what appears 258 to be the real motivation behind the creation of new journals in Open Access: the 259 company’s portfolio interests in the market, rather than the needs of society or the 260 pursuit of scientific innovation.”

Deleted.

Line 269: Also see: Shen, C., & Björk, B.-C. (2015). ‘Predatory’ open access: a longitudinal study

of article volumes and market characteristics. BMC Medicine, 13(230), 1-15. doi:

10.1186/s12916-015-0469-2

You should expand the discussion of predatory OA and how DOAJ hopes to provide a “white list” that eliminates those.

We have added this reference beneath a previous one (Somoza et al, 2018). We have included also a comment about the role of DOAJ against predatory journals (“white list”).

Since in your conclusion (lines 283-285) you make a bold prediction (283: “The high concentration of journals controlled by commercial publishers may generate a price crisis in the Open Access movement similar to that of the 1960s, when prices rose so high that libraries were forced to cancel subscriptions.”) you need to include analysis on prices. This statement again needs evidence and your study is in a position to do so, or at least include references and a discussion of price studies by others.

We have changed the sentence and the argument and we have added the reference to Khoo, S.Y.-S., 2019. 

In summary: include characteristics of publishers in introductory section, remove or substantiate several statements that are not substantiated by evidence from data analysis or literature review, add to the literature and discussion on predatory journals and how DOAJ serves as a white list to help authors select reputable OA journals.

Ok, thanks for your comments.

Reviewer #2: 

This manuscript represents an interesting analysis of the publishers business model and journals' behaviour respect to some critical variables in the OA era. Focusing on the DOAJ seal is a good point of analysis. The overall manuscript is well-written, the aims of the study and the conclusions are consistent. Just a few observations:

2.Methodology section: when you give numbers, please specify the date of your search as nowaday the numbers are different, of course, even if slightly.

Changed (line 127 … “in March of 2019”.)

3.1: the observation of lines 210-212 deserves to be stressed also in the conclusion section because is very meaningful!

We have added a comment in the conclusions (line 440)

3.2 Distribution of knowledge areas, line 257-260: this statement is more a consideration than a description. you could put it in the Results and Discussion section.

We agree on that, but this statement is now in the Results and Discussion section. This consideration or reflection is based on the data collected and starts a relevant discussion.

Just a final observation: discussing about PLOSone in PLOSone...I think your analysis is quite objective but... in conclusion, it's a manuscript, very interesting to be read!

Thanks for your comments. We have tried to discuss about open access publishers and PLOSOne is part of this reality. We think also that it is a good journal to send our manuscript and for improving it (correcting any bias or problem) with the help of our reviewers.

---

## [Decision Letter · Decision Letter 1]

23 Apr 2020

PONE-D-20-00381R1

Open access publishers: the new players

PLOS ONE

Dear Dr Abadal,

Thank you for submitting your manuscript to PLOS ONE. After careful consideration, we feel that it has merit but does not fully meet PLOS ONE’s publication criteria as it currently stands. Therefore, we invite you to submit a revised version of the manuscript that addresses the points raised during the review process.

Firstly, I would like to acknowledge the pertinence and quality of the amendments and clarifications, as well as the responses provided to our two reviewers. Both have expressed their satisfaction in regard to the work done during the revisions of your paper. You will find below (please see the comments from Reviewer # 2) an additional minor comment that, I believe, requires your attention and could maximize the value of the paper. Once received your revised manuscript, I will assess it and immediately proceed to make an editorial decision without needing an additional round of reviews.

We would appreciate receiving your revised manuscript by Jun 07 2020 11:59PM. To enhance the reproducibility of your results, we recommend that if applicable you deposit your laboratory protocols in protocols.io, where a protocol can be assigned its own identifier (DOI) such that it can be cited independently in the future. For instructions see: http://journals.plos.org/plosone/s/submission-guidelines#loc-laboratory-protocols

We look forward to receiving your revised manuscript.

Kind regards,

Sergio A. Useche, Ph.D.

Academic Editor

PLOS ONE

Reviewers' comments:

Reviewer's Responses to Questions

**Comments to the Author**

1. If the authors have adequately addressed your comments raised in a previous round of review and you feel that this manuscript is now acceptable for publication, you may indicate that here to bypass the “Comments to the Author” section, enter your conflict of interest statement in the “Confidential to Editor” section, and submit your "Accept" recommendation.

Reviewer #1: All comments have been addressed

Reviewer #2: All comments have been addressed

2. Is the manuscript technically sound, and do the data support the conclusions?

Reviewer #1: Yes

Reviewer #2: Yes

3. Has the statistical analysis been performed appropriately and rigorously? 

Reviewer #1: Yes

Reviewer #2: Yes

4. Have the authors made all data underlying the findings in their manuscript fully available?

Reviewer #1: Yes

Reviewer #2: Yes

5. Is the manuscript presented in an intelligible fashion and written in standard English?

Reviewer #1: Yes

Reviewer #2: Yes

6. Review Comments to the Author

Reviewer #1: Thank you for addressing all of the comments. By making these changes the paper is now stronger. This is an interesting paper.

Reviewer #2: Interesting paper, the observations well considered, just some misspelling at lines 89 - 4 Not 5 type of OA -, 148, 154.

The conclusions are consistent with the aims of the study but, at the end, the DOAJ seal seems forgotten...

I would recommend to remind readers that we are in this context...

7. PLOS authors have the option to publish the peer review history of their article (what does this mean?). If published, this will include your full peer review and any attached files.

Reviewer #1: No

Reviewer #2: No

---

## [Author Response · Author response to Decision Letter 1]

29 Apr 2020

Reviewer #1: Thank you for addressing all of the comments. By making these changes the paper is now stronger. This is an interesting paper.

Many thanks for your remarkable comments which have contribute to improve the paper prominently. 

Reviewer #2: Interesting paper, 

Many thanks for your remarkable comments which have contribute to improve the paper prominently.

the observations well considered, just some misspelling at lines 89 - 4 Not 5 type of OA –

Changed.

148, 154.

We’re sorry but we haven’t found the mistakes in these lines. We hope that the copyeditor will find them.

The conclusions are consistent with the aims of the study but, at the end, the DOAJ seal seems forgotten... I would recommend to remind readers that we are in this context...

Changed, we’ve included a remind.

---

## [Editor Report · Decision Letter 2]

6 May 2020

Open access publishers: the new players

PONE-D-20-00381R2

Dear Dr. Abadal,

We are pleased to inform you that your manuscript has been judged scientifically suitable for publication and will be formally accepted for publication once it complies with all outstanding technical requirements.

With kind regards,

Sergio A. Useche, Ph.D.

Academic Editor

PLOS ONE
---

## [Editor Report · Acceptance letter]

13 May 2020

PONE-D-20-00381R2 

Open access publishers: the new players 

Dear Dr. Abadal:

I am pleased to inform you that your manuscript has been deemed suitable for publication in PLOS ONE. Congratulations! Your manuscript is now with our production department. 

With kind regards,

on behalf of

Dr. Sergio A. Useche 

Academic Editor

PLOS ONE